# Synthesis, Structural, and Mechanical Behavior of *β*-Ca_3_(PO_4_)_2_–ZrO_2_ Composites Induced by Elevated Thermal Treatments

**DOI:** 10.3390/ma15082924

**Published:** 2022-04-17

**Authors:** Nandha Kumar Ponnusamy, Hoyeol Lee, Jin Myoung Yoo, Seung Yun Nam

**Affiliations:** 1Industry 4.0 Convergence Bionics Engineering, Pukyong National University, Busan 48513, Korea; nandhabiochem@gmail.com (N.K.P.); hoyeollee11@gmail.com (H.L.); jinmyoungyoo@gmail.com (J.M.Y.); 2Department of Biomedical Engineering, Pukyong National University, Busan 48513, Korea

**Keywords:** *β*-tricalcium phosphate, zirconia, calcium-deficient apatite, composites, phase transition, sintering

## Abstract

Biocompatible *β*-Ca_3_(PO_4_)_2_ and mechanically stable *t-*ZrO_2_ composites are currently being combined to overcome the demerits of the individual components. A series of five composites were synthesized using an aqueous precipitation technique. Their structural and mechanical stability was examined through X-ray diffraction, Rietveld refinement, FTIR, Raman spectroscopy, high-resolution scanning electron microscopy, and nanoindentation. The characterization results confirmed the formation of *β*-Ca_3_(PO_4_)_2_–*t*-ZrO_2_ composites at 1100 °C. Heat treatment above 900 °C resulted in the degradation of the composites because of cationic interdiffusion between Ca^2+^ ions and O^−2^ vacancy in Zr^4+^ ions. Sequential thermal treatments correspond to four different fractional phases: calcium-deficient apatite, *β*-Ca_3_(PO_4_)_2_, *t*-ZrO_2_, and *m*-ZrO_2_. The morphological features confirm in situ synthesis, which reveals abnormal grain growth with voids caused by the upsurge in ZrO_2_ content. The mechanical stability data indicate significant variation in Young’s modulus and hardness throughout the composite.

## 1. Introduction

Bone defects caused by traumas, congenital disabilities, fatigue loading, and sarcomas are severe problems for individuals that significantly reduce their quality of life. The clinicians use autograft (bone donation from the same person) and allograft (cadaver bone) techniques to treat bone defects [1,2]. However, traditional bone grafts have several disadvantages: postoperative infection, disease transmission, insufficient donor sites, morbidity from the donor site, high costs, and limited accessibility. To overcome these disadvantages and the increased demand for bone-graft substitutes, the researchers focus on synthetic bone-graft materials that have the prerequisite of owning excellent biocompatibility and mechanical features [3,4]. Metallic implant materials have been used as artificial implants that involved the usage of stainless steel (316L SS) and titanium alloys (Ti metal and Ti-6Al-4V) for the last decades. However, these metallic implants are prone to high corrosion rates in a hostile human physiological environment, which has led to the development of alternative materials in the form of ceramics [5].

Phosphate-based ceramics are studied for bone-graft substitutes because of their biocompatibility in a hostile environment and their ability to facilitate bone regeneration in host tissues [6,7]. Hydroxyapatite (HA, Ca_10_(PO_4_)_6_(OH)_2_) and tricalcium phosphate (TCP, Ca_3_(PO_4_)_2_) are gaining much attention as bone-graft substitutes. These materials are chemically similar to bone tissue, biocompatible, and osteoconductive [8,9,10]. However, phosphate-based ceramics have disadvantages, such as phase degradation during thermal treatments and poor mechanical stability [11,12]. Consequently, many researchers have investigated several methods of reinforcing calcium phosphate using bioinert materials, including zirconia. Incorporating bioinert materials in calcium phosphate increases the mechanical properties, such as hardness, toughness, and the fracture toughness in the composites [13]. Zirconia is one of the most desirable implants because of its high biocompatibility, chemical resistance, hardness, and fracture toughness [14,15]. Pure zirconia exhibits three forms of allotropy during thermal treatment, namely the monoclinic (*m*-ZrO_2_, 25−1170 °C), tetragonal (*t*-ZrO_2_, 1170–2370 °C), and cubic (*c*-ZrO_2_, 2370 °C–melting point) phases. During cooling, the zirconia form can be reverted to its initial phase. Among these polymorphs, tetragonal zirconia (*t*-ZrO_2_) is suitable for hard tissue replacement because of its mechanical stability, that is, its high hardness (H ~ 18 GPa) and Young’s Modulus (E ~ 255 GPa) [16,17].

Several approaches for reinforcement using a biocompatible calcium phosphate matrix and zirconia material have been investigated to improve bone regeneration and mechanical stability. From the recent report, Cao et al. [18] prepared a mixture of HA and 3YSZ sourcing by a flash sintering process to elude the phase degradation and promote the densification for mechanical stability. The presence of 60% 3YSZ content and the gradual decomposition of HA lead to pores, lack in densification, uneven grains, and secondary formations. From the same group, Cao et al. [13] investigated HA-ZrO_2_ scaffolds; during the sintering process, a lack of phase stability and the scaffold embedded in stimulated body fluids in the resultant 10 wt% shows better compressive strength. Shaianlou et al. [19] established incorporating graphene oxide (GO), HA, and ZrO_2_ in scaffold fabrication followed by a simulated body-fluid solution to strengthen the composites mixture. The compressive strength shows a reductant incomparable with GO/ZrO_2_ composites. However, during thermal treatments, the preliminary HA and ZrO_2_ mixture undergoes phase degradation, which leads to the formation of a secondary phase, including *β*-Ca_3_(PO_4_)_2_, CaO, CaZrO_3_, Ca_4_(PO_4_)_2_O, and α-Ca_3_(PO_4_)_2_ [20,21,22]. Furthermore, the secondary phase grains are impregnated on the lattice site of the composites, thus degrading the mechanical stability [23,24]. Considering the drawback of HA, bioactive tricalcium phosphate can be an alternative substitute because of its excellent biocompatibility and osteoconductive properties. The combination of inert materials such as *β*-Ca_3_(PO_4_)_2_–TiO_2_ and *β*-Ca_3_(PO_4_)_2_–Al_2_O_3_ have recently been reported to improve the structural stability by elevating temperatures. Composites with stable phases have garnered attention because of their exceptional mechanical stability, antibacterial property, and low friction with living tissues [25,26]. The partially stabilized zirconia’s mechanical strength can be attributed to enhancing the physicochemical property of the composites mixtures.

The current study aims to develop and fabricate the Ca_3_(PO_4_)_2_ and ZrO_2_ composites without using additive agents to improve mechanical and thermal stability. A wide range of various composites was prepared via in situ synthesis methods. The investigation involves detailed structural analysis, followed by investigations of the microstructural and mechanical stability of the Ca_3_(PO_4_)_2_–ZrO_2_ composites.

## 2. Materials and Methods

### 2.1. Powder Synthesis

Simple wet precipitation techniques were utilized for the preparation of Ca_3_(PO_4_)_2_–ZrO_2_ composites with different composition ratios by varying the ZrO_2_ precursor concentration. All the chemicals, including Ca(NO_3_)_2_, (NH_4_)_2_HPO_4_, and ZrOCl_2_, were purchased from Sigma Aldrich (St. Louis, MO, USA), and five composite combinations were formulated for powder synthesis. Pure TCP with a Ca/P ratio of 1.5 for comparative purposes and variations in the ZrO_2_ precursor concentration are represented by samples codes, as shown in Table 1. Powder synthesis was conducted by the addition of the prepared cationic precursors Ca^2+^ and Zr^4+^. This was mixed as the (NH_4_)_2_HPO_4_ anion precursor was slowly added dropwise to the cationic solution; stirring was maintained at 250 RPM and the operating temperature was fixed at 90 °C for the entire reaction. Afterward, the NH_4_OH solution was added to maintain approximately pH 8, with further stirring for 2 h. The precipitate samples were filtered via vacuum filtration and dried in a hot-air oven (TECHNICO OVEN, Technico Laboratory Products Pvt. Ltd., Chennai, India) at 120 °C for 24 h. The precipitate dried powder was crushed with a mortar and pestle to obtain a fine powder. The desiccated composite powder was prepared for heat treatment in a muffle furnace (MATRI-MC 2265 A, Matri Instruments & Chemicals Pvt. Ltd., Puducherry, India) at a heating rate of 5 °C min^−1^ and atmospheric conditions for a 2 h dwell time.

### 2.2. Physiochemical Characterization

The heat-treated Ca_3_(PO_4_)_2_-ZrO_2_ composites were qualitatively analyzed and characterized via X-ray diffraction analysis (Ultima IV, Rigaku, Tokyo, Japan) using Cu Kα radiation (λ = 1.5406 Å) at 40 kV and 30 mA. The diffraction angles (2θ) between 10 and 70° were scanned with a step size of 0.02° 2θ per second. Phase determinations were conducted using the standard International Centre for Diffraction Data (ICDD) Card Nos. 00-009-0169 (*β*-Ca_3_(PO_4_)_2_), 00-009-0432 (Ca_10_(PO_4_)_6_(OH)_2_), 01-079-1765 (*t*-ZrO_2_), and 01-083-0944 (*m-*ZrO_2_). The vibrational modes of the composite powders were determined from the backscattering geometry of the confocal Raman microscope (Renishaw, Gloucestershire, UK). All the composite powder samples were excited at a wavelength of 785 nm by a semiconductor diode laser (0.5% of power) at a data acquiring time of 30 s. Fourier transform infrared spectroscopy (FT-IR) in the transmission mode was performed using an FT-IR spectrophotometer (PerkinElmer, Shelton, CT, USA); in the IR region (4000−400 cm^−1^), it was performed via the KBr method to determine the functional groups in the heat-treated composite powders. The microstructural features of the *β*-Ca_3_(PO_4_)_2_−ZrO_2_ composites were determined through high-resolution scanning electron microscopy (HRSEM; FEG-200, FEI-Quanta, Eindhoven, Netherlands). The procedure described in our earlier reports was utilized in the mechanical study [26,27]. Nanoindentation (CETR, Campbell, CA, USA) was performed using the mechanical data to determine the Young’s modulus and hardness for the 1100 °C heat-treated composites. The schematic representation of the synthesis and fabrication process of the *β*-Ca_3_(PO_4_)_2_–ZrO_2_ composites is shown in Figure 1

Quantitative studies were conducted via Rietveld refinement analysis using GSAS-EXPGUI (V1208, National Laboratory, Los Alamos, NM, USA) software for complete refinement analysis. The standard crystallography information files (CIFs) data were obtained from the American Mineralogist database [28]. The CIF for *β*-Ca_3_(PO_4_)_2_ was obtained from Yashima et al. [29] and a detailed description covering the rhombohedral setting, space groups, and lattice parameters were used for refinement analysis; R3c, Z = 21, a = 10.4352 Å, and c = 37.4029 Å. The atomic arrangement of *β*-Ca_3_(PO_4_)_2_ includes 18 independent atomic positions: 5 Ca positions (3 in site 18b and 2 in site 6a at one-half occupancy), 3 P positions (2 in site18b and 1 in site 6a), and 10 O positions (9 in site 18b and 1 in site 6a). The corresponding CIF data were obtained for HA, *t*-ZrO_2_, and *m*-ZrO_2_ [30,31,32].

## 3. Results

### 3.1. X-Ray Diffraction Analysis

Pure TCP powder with a Ca/P ratio of 1.5 was compared with five different *β*-Ca_3_(PO_4_)_2_–ZrO_2_ composites during the preliminary synthesis. The X-ray diffraction pattern of the powder composites after heat treatment at 700 °C (Figure 2a) was evaluated. A calcium-deficient apatite (CDA) phase was observed in the pure *β*-TCP; a surplus apatite phase and higher crystallization corresponded to an increase in the ZrO_2_ concentration. Generally, the CDA is observed in the *β*-Ca_3_(PO_4_)_2_ at 780 °C. The XRD pattern is in good agreement with the existing standard ICDD reference cards. Furthermore, the pattern observed in the composites at 700 °C was verified by heating to 900 °C.

The XRD reflection pattern reveals that the apatite formed during the *β*-Ca_3_(PO_4_)_2_ phase was observed in the pure TCP and accompanied the initial formation of apatite in the *β*-Ca_3_(PO_4_)_2_ composites. Likewise, crystallization occurs in ZrO_2_ and is observed in the allotropy phase for *t*-ZrO_2_ and *m*-ZrO_2_ at 900 °C. Furthermore, a steady increase in the ZrO_2_ content and a decrease in the apatite phase are clearly observed in the XRD reflection pattern at 900 °C (Figure 2b). The existence of the apatite phase, which is inimical to the crystallization of the *β*-Ca_3_(PO_4_)_2_ phase, was observed in the XRD pattern at 900 °C. Moreover, XRD patterns were recorded for the samples treated at 1000 °C. The reflection patterns reveal the presence of *β*-Ca_3_(PO_4_)_2_. Along with predominant peaks, dual-phase *m-*ZrO_2_, *t-*ZrO_2_, and the apatite phase are observed at 1000 °C (Figure 2c). The XRD reflection pattern at 1100 °C reveals diffracted peaks corresponding to *β*-Ca_3_(PO_4_)_2_, *m-*ZrO_2_, and *t-*ZrO_2_ and an increase in zirconia concentration. The augmentation peak intensity corresponded to *m*-ZrO_2_ and *t*-ZrO_2_, and a decreasing trend is observed for *β*-Ca_3_(PO_4_)_2_. The *β*-Ca_3_(PO_4_)_2_−ZrO_2_ composites revealed the complete removal of the apatite phase at 1100 °C (Figure 2d).

### 3.2. Raman Spectroscopy

The Raman spectra were measured for the *β*-Ca_3_(PO_4_)_2_–ZrO_2_ composite at three different temperatures. The Raman bands were observed at 700 °C (Figure 3a) for the apatite phase, with bands at ~960 cm^−1^ (ν_1_ mode), 430 and 447 cm^−1^ (ν_2_ modes), ~1026 and 1076 cm^−1^ (ν_3_ modes), and 579, 591, and 615 cm^−1^ (ν_4_ modes). Moreover, *β*-Ca_3_(PO_4_)_2_ bands were observed; the sharp intensity and shoulder peaks for symmetric stretching were measured at ~947 and 969 cm^−1^ (ν_1_ mode); bending P-O bands were measured at ~406, 441, and 481 cm^−1^ (ν_2_ modes); stretching P-O bands at 1046 and 1089 cm^−1^ (ν_3_ modes); and triply degenerate asymmetric P-O bending was measured at ~548, 612, and 627 cm^−1^ (ν_4_ modes) [33,34]. In addition, peaks emerged for the *t-*ZrO_2_ peaks at 145, 222, 263, 381, 473, and 637 cm^−1^, and *m*-ZrO_2_ peaks were observed at 178 and 189 cm^−1^, corresponding to the Zr-O vibrations modes [35]. At 700 °C, the Raman spectra revealed the presence of an apatite phase, which was detected at 960 cm^−1^ to represent CDA (Ca/P ratio = 1.5). Moreover, major and minor boosted peaks are observed, which are attributed to the Zr-O vibration band of the dual phase of *t-*ZrO_2_ and *m-*ZrO_2_. The spectra observed at 900 °C demonstrated four different phases: *β*-Ca_3_(PO_4_)_2_, *m-*ZrO_2_, *t-*ZrO_2_, and HA (Figure 3b). The inset reveals that the apatite phase is retained at the 962 cm^−1^ bands; the shoulder peaks of *β*-Ca_3_(PO_4_)_2_ at 945 cm^−1^ revealed a minor shift to a higher wavenumber. Furthermore, a gradual increase in the *t*-ZrO_2_ and *m*-ZrO_2_ peaks corresponded to an increase in the ZrO_2_ precursor concentration. The complete elimination of apatite bands occurred at 1100 °C (Figure 3c). The presence of *β*-Ca_3_(PO_4_)_2_, *m*-ZrO_2_, and *t*-ZrO_2_ at 1100 °C was corroborated.

### 3.3. Fourier-Transform Infrared Spectroscopy 

Infrared spectra were recorded for the *β*-Ca_3_(PO_4_)_2_–ZrO_2_ composites at 900 °C (Figure 4a) and 1100 °C (Figure 4b). The infrared spectra of *β*-Ca_3_(PO_4_)_2_ for the P-O groups are prominent at ~943 and 973 cm^−1^ for symmetric P-O stretching (ν_1_), ~430 cm^−1^ for doubly degenerate O−P−O bending (ν_2_), ~1192 and 1206 cm^−1^ for triply degenerate asymmetric P-O stretching (ν_3_), and ~555 and ~610 cm^−1^ for triply degenerate O-P-O bending (ν_4_).

Infrared apatite spectra are observed at ~960 cm^−1^ for symmetric P−O stretching (ν_1_); with regard to the OH group, they are observed from ~3200 to 3800 cm^−1^, which corresponds to the HA peaks. The PO_4_ tetrahedron groups correlate with the characteristic apatite peaks and the *β*-Ca_3_(PO_4_)_2_ phase at 900 °C. Regarding the IR spectra, major peaks related to the hydroxyl group are observed at ~3439 cm^−1^ and 900 °C [36]. Moreover, sharp peaks for the Zr-O group in the monoclinic zirconia are observed at 743 cm^−1^, corresponding to an increase in the concentration of zirconia precursor and a gradual increase in peak intensity [37]. Regarding the FTIR spectra recorded at 1100 °C, the apatite phase for the OH group is eliminated. However, a negligible change in the apatite phase is observed between 900 °C and 1100 °C, and a strong intensity peak is observed for the monoclinic zirconia at 1100 °C. Generally, the X-ray diffraction and Raman spectra corroborated the formation of the *β*-Ca_3_(PO_4_)_2_–ZrO_2_ composites.

### 3.4. Rietveld Refinement

The XRD, Raman, and FT-IR investigations confirm the phase formation of the *β*-Ca_3_(PO_4_)_2_–ZrO_2_ composites following consecutive heat treatments. Structural analysis was performed at 900 °C, 1000 °C, and 1100 °C for the *β*-Ca_3_(PO_4_)_2_–ZrO_2_ composites. The refinement diffraction pattern is presented in (Figure 5), whereas the refined phase composition, Rietveld agreement factors, and lattice parameters are listed in Table 2 and Table 3. The refined 900 °C data confirm the presence of Ca_10_(PO_4_)_6_(OH)_2_, *β*-Ca_3_(PO_4_)_2_, *m*-ZrO_2_, and *t*-ZrO_2_, corresponding to a crystalline Ca_10_(PO_4_)_6_(OH)_2_ hexagonal structure with a space group (P63/m (176)), *β*-Ca_3_(PO_4_)_2_ with a rhombohedral structure (space group-R3c (137)), *m*-ZrO_2_ with a monoclinic structure with a space group (P121/c1), and *t*-ZrO_2_ with a hexagonal structure with a space group (P42/nmc) unit cell, respectively. The refined data agree with the phase formation of *β*-Ca_3_(PO_4_)_2_–ZrO_2_, which was confirmed at 1100 °C.

The phase fractions at 900 °C from the refined data for the five different composites reveal a gradual increase in *t*-ZrO_2_ and *m*-ZrO_2_ content and a simultaneous reduction in *β*-Ca_3_(PO_4_)_2_ and the apatite phase. The refined phase fractions at 1000 °C at 1CPZ and 2CPZ demonstrate a negligible apatite phase and a significant and steady increase in the *β*-Ca_3_(PO_4_)_2_ phase fraction. The apatite phase is evident at 3CPZ, 4CPZ, and 5CPZ, which displays an increase in the phase fraction and a decrease in the *β*-Ca_3_(PO_4_)_2_ phase. However, the *t*-ZrO_2_ and *m*-ZrO_2_ phases are dominant in the composites. In the comparative phase composition of *β*-Ca_3_(PO_4_)_2_, *t*-ZrO_2_, and *m*-ZrO_2_, the complete elimination of the apatite phase is observed at 1100 °C. The data of the refined lattice parameters reveal the refined phase fraction at 900 °C and 1100 °C. A considerable contraction in the lattice parameter of *β*-Ca_3_(PO_4_)_2_, demonstrated in *a = b axis* and the *c-axis*, reveals marked differences in reductance compared with pure TCP; the decrease in the lattice parameters corresponds to an increase in ZrO_2_ content at 900 °C and 1100 °C. The preferential occupancy factors that were refined at 1100 °C are listed in Table 4, which demonstrated that Zr^4+^ was accommodated at the Ca5 sites of the *β*-Ca_3_(PO_4_)_2_ lattice throughout the composites. In contrast, the *t-*ZrO_2_ and *m-*ZrO_2_ lattice parameters exhibit no significant changes following an increase in the ZrO_2_ phase contents.

### 3.5. Morphological and Mechanical Features

The morphological characteristics of the *β*-Ca_3_(PO_4_)_2_–ZrO_2_ composite pellets heat treated at 1100 °C are indicated in (Figure 6). The grains in 1CPZ were randomly distributed throughout the microstructure and numerous voids were present. The grain growth in 5CPZ is haphazard and the grains are overlaid with pores; this indicates that their growth commenced at 1100 °C. The increase in grain distribution within the microstructure corresponds to an increase in the ZrO_2_ concentration.

Moreover, energy-dispersive X-ray spectroscopy indicates that the elemental composition is dominated by Ca, Zr, P, and O. The element distribution in 1CPZ and 5CPZ is in good agreement with the refined phase fractions. The mechanical stability data from the nanoindentation test are listed in Table 4 and the load displacement profiles are shown in Figure 7. The loading and unloading profiles in all the compositions are irregular. Micropores and twining grains are evident in the microstructures of the composites, and it evinces the existence of the *t-*→ *m*-ZrO_2_ toughening mechanism. The indentation profiles validate the stabilized zirconia grains’ smooth loading and unloading profiles and an inadequate profile for the porous structure. However, the mechanical data reveal that Young’s modulus and hardness decrease with an increase in the ZrO_2_ precursor concentrations.

## 4. Discussion

The results revealed the formation of a stable phase following the formation of the *β*-Ca_3_(PO_4_)_2_–ZrO_2_ composite mixtures with varying Zr concentrations through the wet precipitation method. The crystallization of ZrO_2_ for the *t-*ZrO_2_ and CDA phases commenced at 700 °C. The conversion of the CDA (Ca/P ratio = 1.5) to the *β*-Ca_3_(PO_4_)_2_ phase occurred at ~780 °C, as reported earlier [38,39]. Numerous cation materials have recently been investigated for the *β*-Ca_3_(PO_4_)_2_ composites and varied outcomes have been reported [40,41]. The phase changes for the individual cation, which delays the conversion of CDA to *β*-Ca_3_(PO_4_)_2_. Due to the unique cation physiognomies, phase conversion occurs in Al_2_O_3_, TiO_2_, and Fe_2_O_3_; ZnO, CeO_2_, and Y_2_O_3_ delay the apatite transformation [42]. In contrast, *β*-Ca_3_(PO_4_)_2_ with zirconia-composite additives demonstrates the influence of the Dy^3+^, Y^3+^, and Gd^3+^ stabilizers, which delay the apatite phase transformation following an increase in temperature [43,44]. Furthermore, *t*-ZrO_2_ and *m*-ZrO_2_ are observed throughout the composites. The *t*-ZrO_2_ phase stabilization is validated by (1) the moiety of a hydroxyl group and absorbed water molecule, which prompts the *t-*ZrO_2_ phase at low-temperature sintering, and (2) the fact that Ca^2+^ ions may be partially cleaved in the *β*-Ca_3_(PO_4_)_2_ lattice site. The partially cleaved Ca^2+^ ions were accommodated in the Zr^4+^ lattice sites, which emerged to form the *t*-ZrO_2_ phase. The scarcity of Ca^2+^ ions is attributed to the *m*-ZrO_2_ phase transformation [45,46].

The phase changes for the individual cation, which delays the conversion of CDA to *β*-Ca_3_(PO_4_)_2_. Due to the unique cation physiognomies, phase conversion occurs in Al_2_O_3_, TiO_2_, and Fe_2_O_3_; ZnO, CeO_2_, and Y_2_O_3_ delay the apatite transformation. In contrast, *β*-Ca_3_(PO_4_)_2_ with zirconia-composite additives demonstrates the influence of the Dy^3+^, Y^3+^, and Gd^3+^ stabilizers, which delay the apatite phase transformation following an increase in temperature. Furthermore, *t*-ZrO_2_ and *m*-ZrO_2_ are observed throughout the composites. The *t*-ZrO_2_ phase stabilization is validated by (1) the moiety of a hydroxyl group and absorbed water molecule, which prompts the *t-*ZrO_2_ phase at low-temperature sintering, and (2) the fact that Ca^2+^ ions may be partially cleaved in the *β*-Ca_3_(PO_4_)_2_ lattice site. The partially cleaved Ca^2+^ ions were accommodated in the Zr^4+^ lattice sites, which emerged to form the *t*-ZrO_2_ phase. The scarcity of Ca^2+^ ions is attributed to the *m*-ZrO_2_ phase transformation.

The phase fraction characterizes the *β*-Ca_3_(PO_4_)_2_–ZrO_2_ composites at the three different temperatures, demonstrating the decomposition of CDA following sequential heat treatments. Consequently, a steady increase in the *t*-ZrO_2_ and *m*-ZrO_2_ phases and a decrease in *β*-Ca_3_(PO_4_)_2_ were observed. The Raman spectra in Figure 3 indicate a potential deterioration in the apatite phase following thermal treatment of the composites. The gradual increase in *t*-ZrO_2_ and *m*-ZrO_2_ in the composite mixtures is evident. Extensive cation inclusion and substitution, from monovalent to tetravalent cations, in the *β*-Ca_3_(PO_4_)_2_ lattice reportedly result in thermal stability and retained biological functions [47,48]. Depending on the valence effect and ionic size, small and large cations replace the Ca^2+^(sites). The occupancy factor of Zr^4+^ (0.79 Å) at the Ca^2+^ (1.00 Å) site accommodated the Ca^2+^ (5) locations in the *β*-Ca_3_(PO_4_)_2_ lattice. However, a minimum amount of Zr^4+^ can occupy the Ca^2+^ (5) sites and the excess Zr^4+^ ions form the ZrO_2_ composites. However, the lattice parameters exhibit consistent contractions in the *β*-Ca_3_(PO_4_)_2_ lattice sites, which affect the incorporation of the Zr^4+^ ions. The Ca^2+^ ions are deficient in the *β*-Ca_3_(PO_4_)_2_ structure; their action on the Zr^4+^ ions leads to crystallization and the formation of *t-*ZrO_2_. The twofold mixture of the *t*-ZrO_2_ and *m*-ZrO_2_ phases is achieved by the addition of CaO (from 0.5 to 1.2 mol%), and the stable *t*-ZrO_2_ phase is obtained by the addition of more than 9.3 mol% CaO [49]. Moreover, the results show the lack of a secondary phase formation. This confirms a CaO deficiency, which prevents the formation of CaZrO_3_ as an end product.

The morphology of *β*-Ca_3_(PO_4_)_2_–ZrO_2_ is irregularly arranged grains, with numerous voids present in 1CPZ and 5CPZ, which results in poor mechanical stability. The uneven grains and pores in the microstructure are due to the thermal expansion of individual grains in the composite mixtures. The hardness and Young’s modulus from the nanoindentation data exhibit a decrease following the gradual increase in precursor concentration. The drastic decrease in elastic modulus and hardness with the increasing porosity of the composites suggests that binding energy has been released. This indicates that increasing the density and toughness is required to constrain grain growth, thus limiting the *t*-ZrO_2_ → *m*-ZrO_2_ phase change following thermal treatment. However, the hardness and Young’s modulus of the *β*-Ca_3_(PO_4_)_2_–ZrO_2_ suggest better mechanical stability than natural bone [50].

## 5. Conclusions

A simple co-precipitation approach presents the formation of the *β*-Ca_3_(PO_4_)_2_–ZrO_2_ composites with varying compositional ratios. The phase transition from CDA to pure *β*-Ca_3_(PO_4_)_2_ occurs at 1100 °C. The compositional ratios comprise a wide range of Ca^2+^ and Zr^4+^ precursor concentrations. Zr^4+^ is preferentially accommodated at the *β*-Ca_3_(PO_4_)_2_ crystal structure and its continuous accumulation leads to the formation of ZrO_2_. Adding zirconia beyond the saturation limit of *β*-Ca_3_(PO_4_)_2_ led to the crystallization of *m*-ZrO_2_ and *t*-ZrO_2_, and further additions resulted in a sharp increase in the ZrO content. The surface morphological features include irregular grains with numerous voids, and Ca^2+^, P^+5^, Zr^4+^, and O^2−^ are uniformly distributed throughout the composites. Tests on the mechanical stability confirm the deterioration in Young’s modulus and hardness following augmentation of the ZrO_2_ content.

## Figures and Tables

**Figure 1 materials-15-02924-f001:**
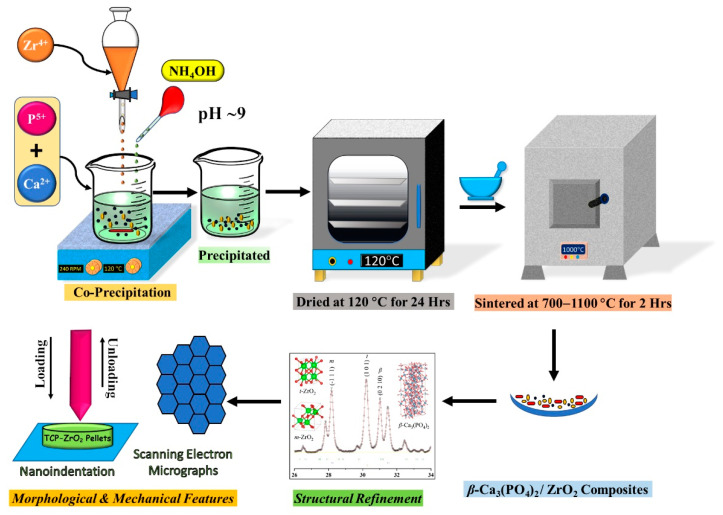
Schematic illustration of the synthesis and fabrication of the *β*-Ca_3_(PO_4_)_2_–ZrO_2_ composites.

**Figure 2 materials-15-02924-f002:**
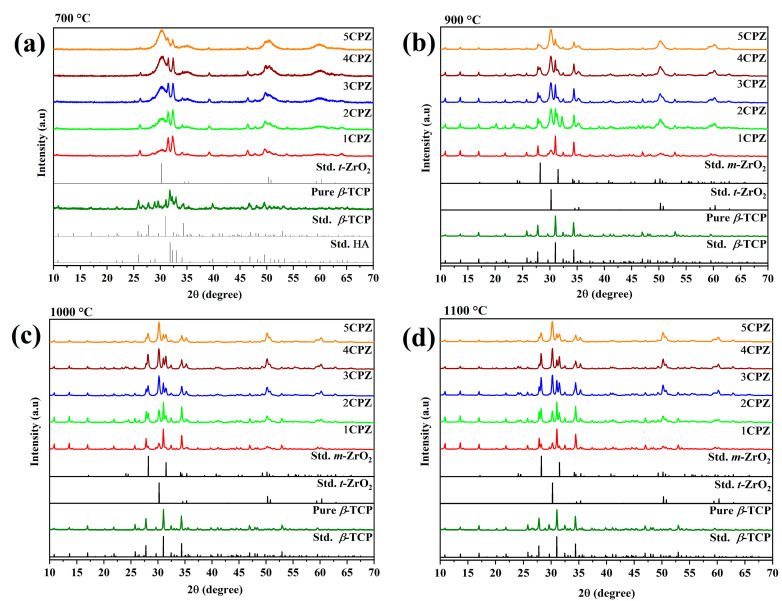
The figures represent the X-ray diffraction of *β*-Ca_3_(PO_4_)_2_–ZrO_2_ composites heat-treated at (**a**) 700 °C, (**b**) 900 °C, (**c**) 1000 °C, and (**d**) 1100 °C, respectively. The diffraction standard for HA-Ca_10_(PO_4_)_6_(OH)_2_, *β*- Ca_3_(PO_4_)_2_, *t*-ZrO2, and *m*-ZrO2 is represented by ICDD Card Nos: 00–009–0432, 00–009–0169, 01–079–1765, and 00–083–0944, respectively.

**Figure 3 materials-15-02924-f003:**
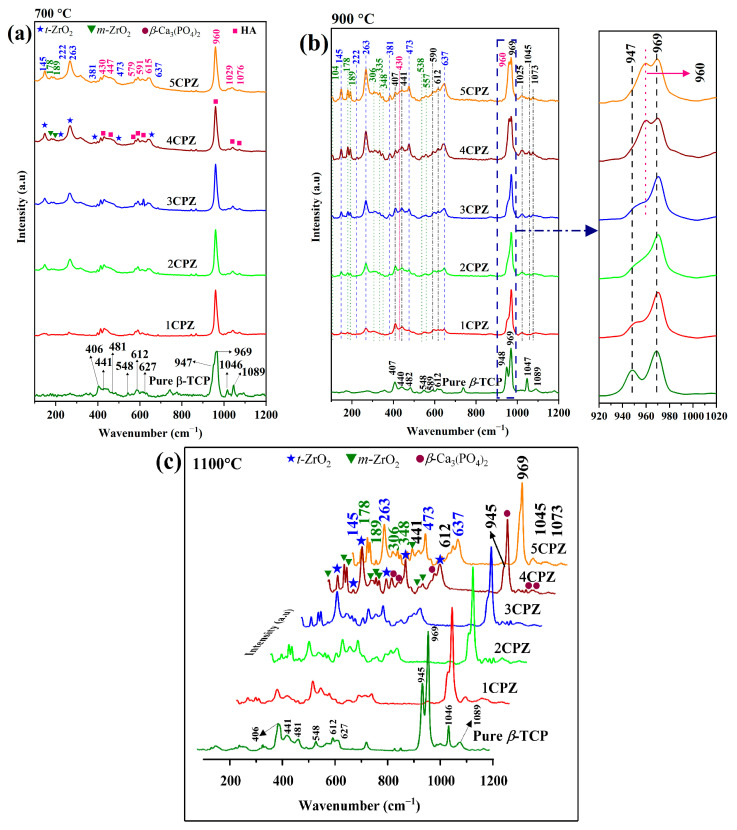
Raman spectra of the *β*-Ca_3_(PO_4_)_2_–ZrO_2_ composites after heat treatment at (**a**) 700 °C, (**b**) 900 °C, and (**c**) 1100 °C.

**Figure 4 materials-15-02924-f004:**
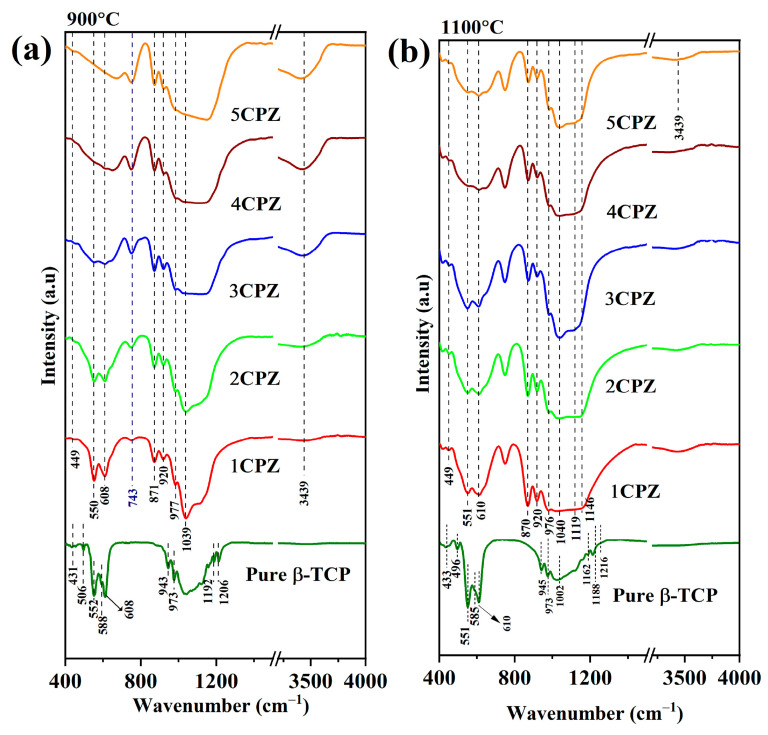
FT-IR spectra of the *β*-Ca_3_(PO_4_)_2_–ZrO_2_ composites after heat treatment at (**a**) 900 °C and (**b**) 1100 °C.

**Figure 5 materials-15-02924-f005:**
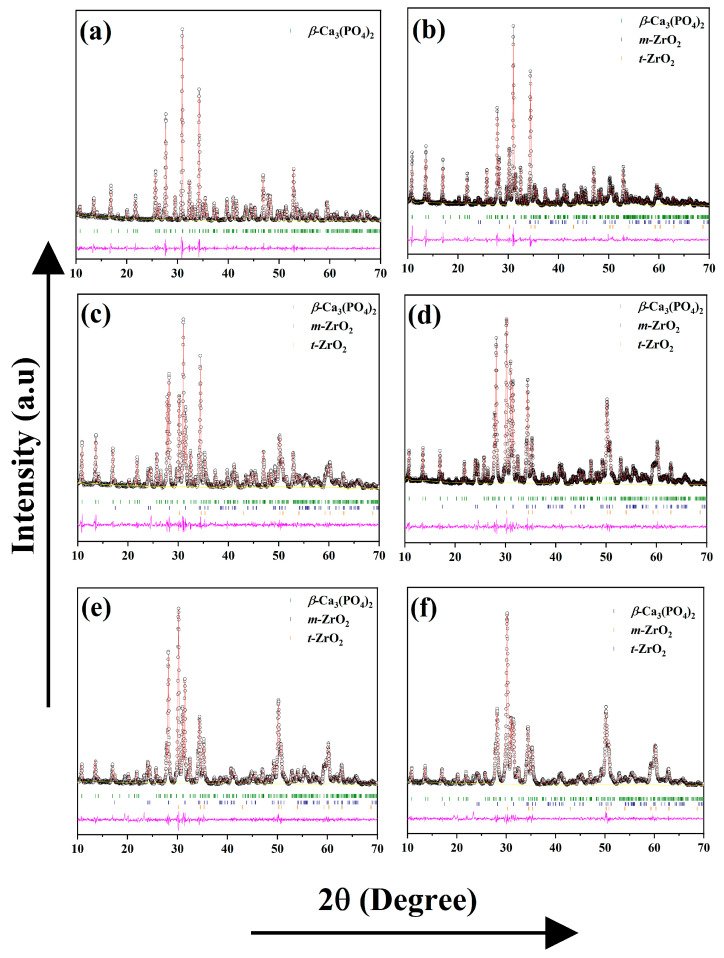
Rietveld refinement diffraction patterns for *β*-Ca_3_(PO_4_)_2_–ZrO_2_ composites at 1100 °C. (**a**) Pure TCP, (**b**) 1CPZ, (**c**) 2CPZ, (**d**) 3CPZ, (**e**) 4CPZ, and (**f**) 5CPZ. The calculated (red line), background (yellow line), and difference (magenta), as well as the Braggs of *β*-Ca_3_(PO_4_)_2_ (olive lines), *t-*ZrO_2_ (orange lines), and *m-*ZrO_2_ (navy lines).

**Figure 6 materials-15-02924-f006:**
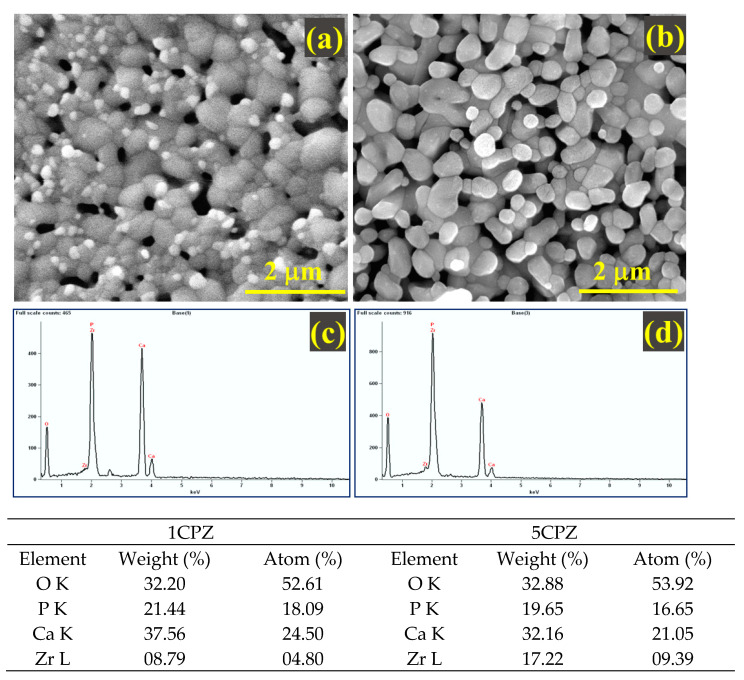
Microstructural features and energy-dispersive X-ray spectra of the *β*-Ca_3_(PO_4_)_2_–ZrO_2_ composites sintered at 1100 °C. (**a**,**b**) 1CPZ and 5CPZ micrographs, (**c**,**d**) corresponding EDS maps reveal the elemental composition: O, P, Ca, and Zr of 1CPZ and 5CPZ, respectively.

**Figure 7 materials-15-02924-f007:**
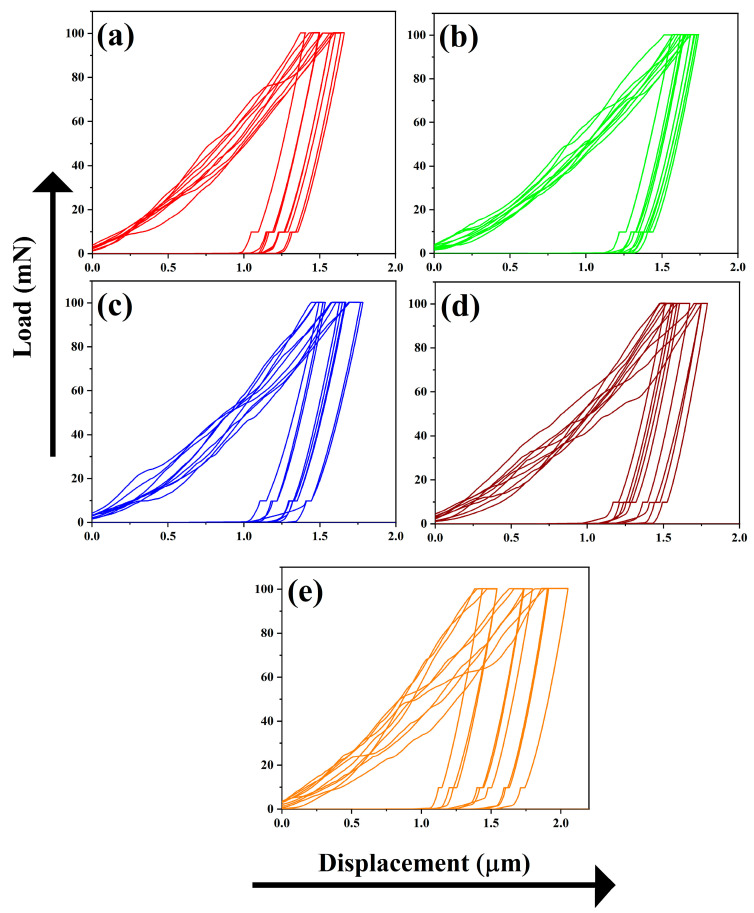
Nanoindentation test representing the five different *β*-Ca_3_(PO_4_)_2_–ZrO_2_ composites sintered at 1100 °C. (**a**) 1CPZ, (**b**) 2CPZ, (**c**) 3CPZ, (**d**) 4CPZ, and (**e**) 5CPZ.

**Table 1 materials-15-02924-t001:** Chemical composition of the prepared *β*-Ca_3_(PO_4_)_2_–ZrO_2_ composites.

Precursor Concentration in a Molar Ratio (mol L^−1^)					
Sample Code	Ca(NO_3_)_2_∙4H_2_O	NH_4_H_2_PO_4_	ZrOCl_2_∙8H_2_O	Ca/P	(Ca + Zr)/P
Pure TCP	1.0 M	0.6667	-	1.50	1.500
1CPZ	1.0 M	0.6667	0.2	1.50	1.800
2CPZ	1.0 M	0.6667	0.4	1.50	2.100
3CPZ	1.0 M	0.6667	0.6	1.50	2.400
4CPZ	1.0 M	0.6667	0.8	1.50	2.700
5CPZ	1.0 M	0.6667	1.0	1.50	3.000

**Table 2 materials-15-02924-t002:** Rietveld refinement agreement factors and phase composition data from the XRD pattern for three different temperatures.

Sample Code		Mineralogical Composition (Wt%)				Refinement Parameters			
		Ca_10_(PO_4_)_6_(OH)_2_	*β*-Ca_3_(PO_4_)_2_	*m*-ZrO_2_	*t*-ZrO_2_	R_wp_	R_p_	χ^2^	R_Bragg_
900 °C	1CPZ	12.67 (2)	73.95 (5)	05.30 (1)	08.08 (2)	10.95	08.51	02.12	05.13
	2CPZ	14.58 (3)	57.86 (2)	11.28 (2)	16.28 (2)	10.58	07.89	01.51	04.64
	3CPZ	20.16 (4)	44.16 (2)	15.77 (2)	19.91 (4)	12.08	08.81	02.17	04.90
	4CPZ	25.87 (6)	29.96 (6)	20.22 (2)	23.95 (3)	12.78	09.18	02.53	05.41
	5CPZ	31.47 (8)	18.50 (1)	24.32 (8)	28.03 (9)	11.03	08.43	01.91	05.28
1000 °C	1CPZ	-	83.78 (1)	06.47 (1)	09.75 (6)	11.88	09.12	02.47	06.01
	2CPZ	-	69.31 (5)	12.25 (3)	18.44 (2)	10.91	08.45	02.38	06.16
	3CPZ	06.33 (2)	55.16 (2)	18.38 (1)	20.13 (2)	10.11	07.83	02.20	03.64
	4CPZ	11.27 (3)	39.46 (3)	22.40 (2)	26.87 (2)	10.80	08.18	02.68	05.16
	5CPZ	14.43 (6)	27.98 (2)	26.01 (1)	31.58 (2)	09.93	07.82	2.319	05.48
1100 °C	Pure TCP	-	100	-	-	07.15	08.44	1.114	04.29
	1CPZ	-	79.30 (3)	07.74 (2)	12.96 (2)	11.72	09.06	02.40	06.09
	2CPZ	-	66.03 (3)	13.68 (2)	20.29 (1)	10.54	08.26	2.236	05.20
	3CPZ	-	59.51 (2)	15.97 (5)	24.51 (2)	08.68	06.76	1.642	05.03
	4CPZ	-	42.49 (2)	19.92 (6)	25.03 (1)	09.84	07.53	02.18	05.71
	5CPZ	-	37.58 (6)	26.81 (4)	35.61 (5)	08.77	06.70	1.855	03.80

**Table 3 materials-15-02924-t003:** Refined lattice parameters for the different phases for the pure TCP and five composites at 900 and 1100 °C.

Sample Code	Refined Lattice Parameter								
	Ca_10_(PO_4_)_6_(OH)_2_		*β*-Ca_3_(PO_4_)_2_		*m*-ZrO_2_			*t*-ZrO_2_	
	*a = b axis*	*c-axis*	*a = b axis*	*c-axis*	*a-axis*	*b-axis*	*c-axis*	*a = b axis*	*c-axis*
	900 °C								
1CPZ	9.5532 (3)	6.8128 (3)	10.4112 (5)	37.4178 (2)	5.1186 (2)	5.1972 (4)	5.2963 (4)	3.5946 (5)	5.1914 (1)
2CPZ	9.5893 (2)	6.8037(3)	10.4086 (8)	37.3946 (3)	5.1460 (2)	5.2041 (2)	5.3156 (2)	3.5934 (4)	5.1891 (8)
3CPZ	9.5924 (8)	6.8199 (1)	10.4101 (1)	37.3925 (4)	5.1504 (2)	5.2047 (1)	5.3159 (2)	3.5924 (5)	5.1909 (1)
4CPZ	9.5636 (5)	6.7571 (6)	10.4015 (2)	37.3617 (7)	5.1496 (2)	5.1927 (1)	5.3159 (1)	3.5900 (5)	5.1851 (1)
5CPZ	9.5465 (5)	6.7788 (5)	10.4016 (2)	37.3638 (7)	5.1504 (1)	5.1988 (1)	5.3139 (2)	3.5940 (4)	5.1817 (9)
	1100 °C								
Pure TCP	-	-	10.4384 (3)	37.4010 (1)	-	-	-	-	-
1CPZ	-	-	10.4203 (4)	37.3877 (1)	5.1481 (8)	5.2092 (9)	5.3124 (8)	3.5948 (3)	5.1868 (8)
2CPZ	-	-	10.4118 (4)	37.3669 (2)	5.1471 (4)	5.2090 (5)	5.3133 (5)	3.5936 (2)	5.1889 (6)
3CPZ	-	-	10.4078 (6)	37.3630 (2)	5.1476 (4)	5.2083 (4)	5.3135 (4)	3.5940 (2)	5.1864 (4)
4CPZ	-	-	10.4063 (7)	37.3600 (3)	5.1452 (4)	5.2075 (5)	5.3118(5)	3.5938(2)	5.1828 (4)
5CPZ	-	-	10.4023 (9)	37.3582 (4)	5.1465 (7)	5.2048 (9)	5.3137 (8)	3.5935 (3)	5.1827 (5)

**Table 4 materials-15-02924-t004:** Refined occupancy factor of Zr^4+^ for Ca^2+^ accommodation in the composites. Young’s modulus and hardness data were acquired from the *β*-Ca_3_(PO_4_)_2_–ZrO_2_ composites at 1100 °C.

Sample Code	Occupancy Factor	Nanoindentation Data	
	Ca^2+^ (5) Sites	Young’s Modulus (GPa)	Hardness (GPa)
1CPZ	0.85 (2)	48.76 ± 02.51	01.79 ± 0.18
2CPZ	0.72 (9)	45.97 ± 01.38	01.59 ± 0.11
3CPZ	0.84 (2)	42.36 ± 02.08	01.51 ± 0.18
4CPZ	0.81 (5)	41.30 ± 04.09	01.36 ± 0.34
5CPZ	0.91 (3)	38.30 ± 02.75	01.30 ± 0.17

## Data Availability

The data that support the findings of this study are available upon reasonable request from the authors.

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
