# Peer review of "Synthesis, Structural, and Mechanical Behavior of β-Ca3(PO4)2–ZrO2 Composites Induced by Elevated Thermal Treatments"

_materials, 2022, doi:10.3390/ma15082924_

Round 1

Reviewer 1 Report

Paper entitled: Synthesis, structural, and mechanical behavior of β-Ca3(PO4)2-ZrO2 composites via progression thermal treatment

For bone applications and the related composites, the authors have introduced five different samples with irregular grains dispersion. To create more stable composites from mechanical-structural aspects, they have used ZrO2 in the composites structure. The examinations were done for different temperatures, 900 oC, 1000 oC and 1100 oC, and loading tests based on Young’s modulus. All in all, the manuscript has a suitable technical formation. So, after removing all the following comments, publication of the manuscript in this journal would be feasible.

  1. Title of the manuscript should be uniform in the capital/ small letters (except propositions like of, the, in and so on).
  2. Please consider a nomenclature list and define all of the variables, parameters and applied abbreviations.
  3. The introduction section should be boosted via reviewing those papers published in the recent three years, 2019-2022.
  4. Preparation stages of the samples (following text of sub-sections 2.1 and 2.2) should be shown in a schematic figure.
  5. Visual quality of figure 4 is not proper. Please present them in a higher quality.
  6. Caption of Figure 5 is not clear. It should be organized so that each figure has a clear description in the common caption.
  7. Using different colors for the curves in figure 6 is not acceptable. Please use different line patterns or symbols to better distinguish the obtained results.
  8. As the authors mentioned, the introduced composite can be available for bone application. What is the reason for testing the composites at a very high temperature, more than 900 oC?
  9. Is it possible to determine the qualification of the composite against the current ones? What about the advantages/ disadvantages of the introduced composites?
  10. If possible, please show the realistic pictures for the proposed compositions. It can be interesting.
  11. The authors have considered the irregular grains structure in their model. Is it possible to have a controlled grains dispersion (regular combination) in the field? A clear response is necessary.  

Author Response

Dear Editor,

My co-authors and I would like to submit the revised version of the manuscript (materials-1648624) "Synthesis, Structural, and Mechanical Behavior of β-Ca3(PO4)2-ZrO2 Composites Induced by Elevated Thermal Treatments” for consideration in Materials. By submitting the revised version, we would like to thank the Editor and Reviewers for their valuable comments that enabled us to improve the quality of the manuscript considerably. Based on the comments, a thorough revision has been done throughout the manuscript and a marked copy (Blue color & Highlighter) of the manuscript has been uploaded as supporting information for review only. A detailed response to the reviewers is attached as a separate document for your perusal.

We are looking forward to hear from you.

Please note that your future correspondence can be addressed to:

Seung Yun Nam, Ph.D.

Associate Professor

Department of Biomedical Engineering,

Pukyong National University

45, Yongso-ro, Nam-Gu, Busan, Korea, 48513,

Reviewer 2 Report

This work deals with the synthesis and characterization of β-TCP/ZrO2 composites. The topic is certainly interesting, although not new. In fact, there is a huge literature about the fabrication and proprieties of hydroxyapatite- or TCP-based materials. In this sense, the introduction section of the manuscript is not appropriate to clearly represent the state-of-the-art of this subject. The authors report pieces of information or literature results without any attempt to organize them. Just as an example, in the introduction section, the authors confuse and mix synthesis and sintering processese or thermal treatments without any discussion. The description of the cited references appears confused and not properly organized. In addition, it is very hard to understand the originality and the novelty of this work.

The meaning of several sentences throughout the manuscript is incomprehensible, as well as the description of some figures or tables (e.g., Table 2) does not match what they show. Even the title of the manuscript is unclear: what should “progression thermal treatment” be? The manuscript is not properly written and organized, and many typos appear in the text.

Accordingly, I suggest not to accept this manuscript for publication.

Author Response

(The authors gave the same response as above.)

Reviewer 3 Report

The submitted paper deals with the synthesis of β-Ca3(PO4)2-ZrO2 composites and their structural and mechanical characterization. It is interesting, complete and well written. In my opinion, it is worth publishing in Materials after minor corrections. My reservations concern the applied codes for these newly obtained composites. It is difficult to relate them to chemical composition. Moreover, it would be a good idea if, having introduced a given abbreviation, the authors consistently used it further on in the study.

Author Response

(The authors gave the same response as above.)

Round 2

Reviewer 1 Report

The authors answered the comments precisely and thus the paper can be accepted in its current form.

Reviewer 2 Report

The revised manuscript can be accepted for publication.